# Haptic Perception Training Programs on Fine Motor Control in Adolescents with Developmental Coordination Disorder: A Preliminary Study

**DOI:** 10.3390/jcm11164755

**Published:** 2022-08-15

**Authors:** Yee-Pay Wuang, Chien-Ling Huang, Ching-Shan Wu

**Affiliations:** 1Department of Occupational Therapy, Kaohsiung Medical University, Kaohsiung 807, Taiwan; 2Department of Rehabilitation Medicine, Chung-Ho Memorial Hospital, Kaohsiung Medical University, Kaohsiung 807, Taiwan

**Keywords:** haptic perception, fine motor control, developmental coordination disorder

## Abstract

Somatosensory and haptic perception deficit was commonly found in developmental coordination disorder (DCD) and was closely related to fine motor functions, and the intervention strategies should thus emphasize improving the underlying haptic functions. This study was intended to investigate the effects of haptic perception training programs on fine motor functions in adolescents with DCD. A total of 82 DCD participants were assigned to either the haptic perception training program (HTP; *n* = 42, 16 females, mean age = 13.32 ± 2.85 years) or standard occupational therapy (SOT; *n* = 40, 16 females, mean age = 13.41 ± 3.8 years) group. Both groups were given 12-week training, twice a week, 30 min per session. Outcome measures were: Jebsen-Taylor Hand Function Test (JTFHT), Bruininks–Oseretsky Test of Motor Proficiency-Second Edition (BOT-2), Test of Visual-Perceptual Skills- Fourth Edition (TVPS-4), and Vineland Adaptive Behavior Scale -Chinese Version (VABS-C). After the intervention, the HTP group outscored the SOT group on most fine motor control (JTFHT and BOT-2) tasks and all TVPS-4 and VABS-C items. The HTP group had greater pre–post changes on fine motor integration, fine motor precision, manual dexterity, and writing. TVPS-4 reached significant intervention gains on visual spatial relations, visual memory, and visual sequential memory. The haptic perception training programs demonstrated benefits in enhancing fine motor control in adolescents with DCD. It could be used as an easy and effective alternative to hospital or school-based therapy during the pandemic.

## 1. Introduction

Developmental coordination disorder (DCD) is one of the most common developmental disorders in school-aged individuals [1], and the prevalence is about 6% of children aged from 5 to 11 [2]. Although the etiology and causes are unclear, the distinct impairments characterized by DCD are motor control deficits, coordination difficulties, and perceptual-motor dysfunctions [3]. These deficits have long-term impact on adolescents’ daily life, academic performance, and vocational pursuits, and may develop into complex psychosocial problems, although DCD was initially diagnosed based on motor coordination difficulties [4].

Cluster analysis has been used to conduct different DCD subtype studies in recent decades and these studies have reported subtypes differing in both number and characteristics [5]; however, the definite subtype recognized by different studies is the type of fine motor control (e.g., manual dexterity). Motor control mode hypotheses are one of those most commonly proposed to explain fine motor difficulties in DCD. When performing the movement, the DCD individuals rely mostly on visual or external feedback and have difficulties shifting to an open-loop or feedforward mechanism as they mature [6]. In contrast, typically developing individuals use visual regard to guide their movements only at early ages and gradually adopt somatosensation instead for more accurate and efficient motor control as maturation takes place [7].

Somatosensation is indispensable for providing the sensory feedback required to direct movements during task execution [7]. Both somatosensory information and feedback strategies play crucial roles in developing fine motor skills. Haptic perception refers to the process of identifying objects by active touching and manipulation, the procedure combines somatosensory perception of object patterns (e.g., texture, wright, and hardness) and proprioception of hand position and its physical conformation. Typically developing children have acquired the ability to match haptic perception with visual perception during infancy, and have completely developed the refined haptic perception and abilities to recognize and differentiate all dimensions of object properties through active touch until adolescence. The adults demonstrate fast, proficient and organized exploratory behaviors [8]. Somatosensory impairments on the hands curtail useful tactile and proprioceptive feedback, causing an individual to be much slower and less efficient in performing most fine motor tasks without abundant visual feedback [9]. Consequently, improving haptic perception might improve the overall somatosensory function, thereby further enhancing fine motor control.

According to the ICF-CY [10], health and functioning have interactions with and impacts on activity participations [11]. Improving functional abilities (e.g., fine motor control) that could enhance activities, engagement, and performance has long been the primary intervention objective for occupational therapists specializing in serving DCD populations. Most occupational therapists have used neurodevelopmental or motor learning practice models to guide interventions to improve fine motor functions; the corresponding treatment protocols emphasize using specific handling techniques and repetitive motor-based activities to enhance the related musculoskeletal and neuromotor functions. These practice models were designed and particularly beneficial for motor difficulties caused by neuromuscular disorders [12]. Previous research has supported the somatosensory and haptic perception deficit in DCD and its close relation with fine motor functions [13], and the intervention strategies should thus emphasize improving the underlying somatosensory and related functions. The present study’s aim was to investigate the efficacy of age-appropriate haptic perception training programs on improving fine motor control functions in DCD adolescents.

## 2. Materials and Methods

### 2.1. Study Design

A quasi-experimental design was utilized in this preliminary study to investigate the effectiveness of a haptic perception training program on fine motor functions and participations in adolescents with DCD.

### 2.2. Participants

The Institute Review Board of Kaohsiung Medical University Hospital approved this study (Protocol Code: KMUHIRB-SV (II)-20190087, Date of approval: 1 August 2019). Written informed assent/consent was obtained from the participants and their parents for enrollment and publication. Different assessments might not generate equivalent results in diagnosing DCD due to their heterogeneous nature; therefore, a three-step sampling procedure was adopted for efficiently identifying DCD individuals [14]. Participants aged from 13- to 16-year-olds were recruited from 28 high schools across different geographic areas. The Developmental Coordination Disorder Questionnaire 2007 (DCDQ’07) [15] and a simplified user manual were provided to the participating parents. A total of 878 questionnaires were sent, and 367 (41.8%) were returned completed. The outcome was that 111 participants with DCDQ’07 scored below 55 were chosen for further motor evaluations, each of them was tested with the Bruininks-Oseretsky Test of Motor Proficiency- 2nd Edition (BOT-2) [16] and Age Band 3 of the Movement Assessment Battery for Children-Second Edition (MABC-2) [17]. Participants were diagnosed as having DCD if they met the following scoring criteria: ≤the 16th percentile on the MABC-2, or ≤the 15th percentile on the BOT-2 Fine Manual Control, Manual Coordination, Body Coordination, Strength and Agility Composite or the Total Motor Composite standard score [14].

### 2.3. Measures

#### 2.3.1. Jebsen-Taylor Hand Function Test (JTHFT) [18]

The JTHFT is a quick and reliable measure of hand-functioning incorporating 7 simulated tasks of activities of daily living (ADL). The tasks were: stacking, simulated page-turning, simulated feeding, picking up small objects, lifting a large and lightweight object, lifting a heavy object, and writing (copying). All tasks were timed and three scores were generated. A JTHFT- object score was the total time (seconds) taken to perform all 6 tasks except for writing for dominant and non-dominant hand, respectively. A JTHFT- writing score was the time spent (seconds) to complete a writing task for the dominant hand. Excellent reliabilities was reported in adults and typically developing children, and this assessment has also been utilized to exam the uni-manual capabilities in children with cerebral palsy and other motor dysfunctions [19].

#### 2.3.2. Bruininks–Oseretsky Test of Motor Proficiency-Second Edition (BOT-2)

The BOT-2 assesses motor proficiencies in individuals aged 4–21 years, the four motor areas include fine manual control, manual coordination, body coordination, and strength and agility. We used the subtests of fine manual control and manual coordination composites to evaluate the intervention effectiveness in the present study. The fine manual control composite evaluates the motor skills involved in tasks requiring accurate hand and finger movements (e.g., copying and drawing), and the manual coordination composite assesses motor skills needed for activities demanding dexterity, speed, and upper extremity coordination (e.g., ball catching and stringing). The BOT-2 has sound reliabilities and validities as well as high responsiveness in children with various disorders, and could be utilized to diagnose DCD and measure their motor functions [20].

#### 2.3.3. Test of Visual-Perceptual Skills—Third Edition (TVPS-4) [21]

The TVPS-4 is a motor-free and individually administered test to evaluate the comprehensive visual perceptual functions for individuals aged 5–21. The TVPS-4 comprises 7 tests: visual discrimination, visual memory, spatial relationship, form constancy, sequential memory, visual figure- ground, and visual closure. The TVPS-4 has excellent reliabilities evidence and concurrent validities with other similar tests [21].

#### 2.3.4. Vineland Adaptive Behavior Scale-Chinese Version (VABS-C) [22]

The VABS-C was the Chinese version of the Vineland Adaptive Behavior Scale [23] and was used in individuals aged 0 to 18 years-11 months. This assessment measures age-related adaptive skills essential for personal and social competencies. This 577-item measure includes four domains: motor skills (gross, fine), communication (written, expressive, receptive,), daily living skills (community domestic, personal,), and socialization skills (coping skills, interpersonal relationships, play and leisure time). The VABS-C has good reliabilities, current, and discriminative validities [22].

### 2.4. Procedures

A total of 88 children (50 males and 38 females; mean age = 13.11, SD = 3.18 years) met the DCD diagnosis criteria after the 3-step screening process. All participants were at mainstream high schools, and their IQ tests, measured at the entrance of high schools, affirmed the absence of intellectual or cognitive impairment (the WISC-VI full IQ of all participants were between 103 and 123). All 88 children were randomly assigned to two equal-sized intervention (*n* = 44) groups by using a computer- generated random table at first, however, nineteen participants (21.6%) asked to be transferred to a different intervention group due to their conflicting schedules.

Each intervention group was treated with a 30 min session every week for 24 weeks [24,25,26]. Two senior pediatric occupational therapists administered either SOT or HTP on an individual basis according to the designated group. Home or school-based programs were not given to the parents and teachers to avoid practice or other confusing effects. Treatment fidelity was verified by examining 20 synchronous online intervention sessions from 2 participating therapists at the first week and at 12 weeks: ten for each group and time duration. Two pediatric occupational therapists not involved in the current study separately rated the level of the therapists’ adherence to specific treatment approach in accordance with the recommended activities listed in the training manual, using a 10-point scale: 1 (non/irregular) to 10 (regular, always), and the average adherence scores of HTP and SOT interventions across raters and time intervals were above 0.96

Another two occupational therapists, blinded to the participants’ group assignment, conducted the JTHFT, BOT-2 and TVPS-4 to the participants before and after the intervention sessions. The examiners were experienced in correctly administering and scoring all these measures. To avoid possible bias, both the examiners did not reacquaint themselves with the pre-test scores of participants administering the post-test. The participants’ homeroom teachers scored the VABS-C at pre- and post-therapy. The participants were individually tested in a quiet location, and an appropriate amount of breaks were provided during the test to decrease the fatigue effects.

### 2.5. Interventions

#### 2.5.1. Haptic Perception Training Program (HPT)

There were 60 object sets and their matching sets used for the HPT, and examples of them are illustrated in Figure 1. The object sets included three-dimensional objects made from assorted materials including cloth, wood, plastics, metal, etc. These objects sized from about 49 cm^3^ to 196 cm^3^. Matching sets included identical object set items and many other items with similar perceptual properties to those in the object sets. All perceptual dimensions were easily discriminated and fully varied per haptics. Six perceptual dimensions: texture, shape, hardness, size, weight, and contour were tested for each individual, with each object including at least three haptic perceptual dimensions.

Each participant was seated across a table from the therapist who administered the HPT, and was instructed that they were going to play a find-and-matching game. The therapist started the procedure with 2 warm-up items for each perceptual dimension to assure that participants with occluded vision realized the activity requirements. In the warm-up items, participants were first given a sample object in the preferred hand and were asked to feel the particular features of the object. After 10 s, participants were instructed to touch and find the matching objects and choose the one identical to the sample object with the designated perceptual dimensions. Formal training immediately followed and was instructed in the same way as the warm-up item. All the participants were asked to touch and explore the sample item from one object set with occluded vision and were told to be attentive to their perceptual dimensions. After thoroughly feeling the object sets, the participants were directed to find the same objects (e.g., size, texture, and number, contour) from the matching sets hidden in a box. Participants were asked to repeat this task again until a correct choice was made, and continue practising the next object set when all correct items had been chosen in the previous object set.

#### 2.5.2. Standard Occupational Therapy (SOT)

Occupational therapists have designed and implemented an array of therapeutic activities to enhance the fine motor skills and functions in DCD individuals. The primary theoretical frameworks guiding fine motor skills treatment are neurodevelopmental and motor-learning practice models that include neuro-developmental treatment (NDT) and perceptual motor (PM) models [12]. In neuro-developmental therapy, therapists use a variety of hands-on methods to facilitate the movement patterns required for the performance of functional skills in daily life. In perceptual motor learning approaches, the child acquires motor skills through therapist-directed structured activities. Both NDT and PM emphasized and provided massive motor experience, repeated practice of target skills and consistent feedbacks [27]. The research teams designed a standard occupational therapy (SOT) program to compare the effectiveness of the proposed HPT, and the SOT consisted of diverse fine motor training activities including the intervention guidelines of both NDT and PM. The exemplary SOT fine motor therapeutic activities were origami, mosaics, painting, puzzles, design coping and tracing, and various card games. The therapists attended to the participant’s overall positioning and alignment during training activities, and graded the difficulty level of the activities as well.

### 2.6. Data Analysis

This study used SPSS 22 for data analysis. The present study used the following scores of each participant for analysis: JTHFT raw score, BOT-2, TVPS-3 and VABS-C scale scores. The multivariate analysis of variance (MANOVA) was conducted to determine pre-intervention test scores differences using group type as a between-subjects factor.

#### 2.6.1. Inter-Group Analysis

A second MANOVA was used to contrast post-intervention score differences in fine motor control, visual perceptual and school functions between these two groups. The follow-up univariate F-tests were conducted by using Scheffe’s *post hoc* comparisons if the MANOVA demonstrated a significant group effect.

#### 2.6.2. Intra-Group Analysis

The paired *t*-test was used to contrast BOT-2 and JTFHT before and after 24-week treatment in the HTP group to investigate HPT effectiveness. Differences of both TVPS-3 and VABS-C between pre- and post-interventions were analyzed for the HTP group as well. The intervention gains for each group on all measures were also summarized and the difference magnitude was quantified by using effect sizes (ES). According to the statistical analysis used in the present study, ES (d) was calculated as the pre-test score difference (post-test mean − pre-test mean) divided by the standard deviation of the pre-test [28].

## 3. Results

### 3.1. Demographic Data

Six children dropped out due to scheduling conflicts and only 82 participants completed all intervention sessions and evaluations: HTP group (*n* = 42, 26 males and 16 females; mean age = 13.32 years, SD = 2.85), and SOT group (*n* = 40, 24 males and 16 females; mean age = 13.41 years, SD = 3.8). Of the participants, 70 (85.4%) used their right hands as their dominance.

### 3.2. HPT Effectiveness

#### 3.2.1. Group Comparability

The assumption of homogeneity of variance was supported by a non-significant result of the Box’s M test of equality of covariance matrices (Box’s M= 332.14, *p* = 0.66). These two groups had no significant differences on age (SOT: mean = 14.87, SD = 3.61; HTP: mean = 14.63, SD = 2.99; *F* = 1.65, *p* = 0.24) or gender (SOT: 24 males and 17 females; HTP: 21 males and 20 females; χ^2^ = 0.66, *p =* 0.65). Family income and primary caregivers’ years of education were evenly distributed between two groups. No significant differences were noted between these two groups at pre-intervention (JTHFT, BOT-2, TVPS-3, and VABS-C) because the overall MANOVAs for the pre-intervention scores did not reach significant level (all *p >* 0.60).

#### 3.2.2. Differences between HPT and SOT Groups after Intervention

The second MANOVA results demonstrated significant group differences on fine motor control, visual perceptual functioning, and adaptive skills measures (Wilks’ lambda = 0.149, F = 1613.95, *p* < 0.0001, partial η^2^ = 0.99). Follow-up univariate F tests were performed accordingly. Due to the number of univariate analyses conducted, the Bonferroni α level was set at 0.003 (0.05/18) for all following analyses to maintain a family-wise error rate smaller than 0.05. As Table 1 demonstrated, the two groups performed significantly differently across most fine motor control (JTFHT and BOT-2) items, although there were no differences on BOT-2 upper-limb coordination and JTFHT Object non-dominant hand between HTP and SOT groups. The HTP group outscored the SOT group on all TVPS-3 and VABS-C tests.

#### 3.2.3. Within-Group Differences before and after the Intervention

As shown in paired *t*-test results, there were significant differences on fine motor control measures in HTP group before and after interventions except for the JTFHT Object non-dominant hand test (Table 2). TVPS-3 tests demonstrated significant improvements on visual discrimination, visual memory, visual spatial relations, and visual sequential memory.

Table 3 summarizes the calculated ES values for HTP and SOT groups. The Cohen *d* values indicated solid improvements on fine motor control functions since the pre-post difference on fine motor measures in the HTP group exceeded 0.6. The ES for the TVPS-3 visual memory, visual spatial relations and visual sequential memory demonstrated significant improvement gains in the HTP group as well.

## 4. Discussion

Perception and action are interdependent and inextricably intertwined, and this relationship is particularly crucial for developing functional skills for school-aged individuals. Bushnell and Boudreau [29] proposed that particular motor functions and skills are essential for developing haptic perception, and other studies have suggested that haptic perception is also essential for acquiring overall motor functioning [30]. Maturation of both haptics and in-hand manipulation underlies the adolescents’ tool-use abilities for academics, prevocational competencies, and activities of daily living. 

Although mature adolescents can explore objects by haptic perception only and are capable of producing various systematic hand movements by different task goals, previous studies have suggested that DCD individuals fail to obtain the necessary sensory input and lack the required haptic perceptions to carry out functional hand movements [13]. The DCD group performed slower on most hand function tests and daily activities demanding fine dexterity and with greater difficulty [31]. Given the evidence that fine motor control during purposeful activity significantly relies on constant haptic information inflow to lead the direction, force, and accuracy of movement, it could be useful clinically to utilize the HPT to enhance fine motor functions.

Our study results also echoed the effects of haptic perception training program where participants showed significant improvements on fine motor integration, precision, and manual dexterity after receiving HPT compared to conventional occupational therapy. Haptic touch itself can be highly discriminating because it involves the combination of the passive sensation of touch (afferent and cutaneous input) with active exploration (efferent and kinesthetic output) [32]. In our study, the fine motor tasks requiring sensory discrimination such as handwriting on various surfaces revealed the largest intervention gains as well. In addition, the stronger correlations between haptic perception and tasks with small object manipulation should be considered when evaluating and treating individuals with DCD. Fine motor control has been assessed by JTHFT in previous studies, and performance time (speed) was the criterion for assessing the hand functioning level. The use of this sole criterion might not be sufficient because haptic perceptual deficits affect not only the speed of performance but as well as the force, direction and accuracy of movement [33]. The BOT-2 fine motor tasks were used to provide a complete description of fine motor control performance in the current study.

Debate about the impact of tactile stimuli on visual perceptions remains inconclusive despite numerous studies demonstrating that multisensory stimulation could have substantial effects on cognitive processing and visual perceptual functions. Previous studies suggested that tactile stimulation influences visual temporal perception in particular, in that the CNS uses consistencies with tactile cues to evaluate and verify the reliabilities of visual information and accordingly adapt their weights. Maunsell and his colleagues [34] also indicated that in a visual area (V4), neuronal activity can be affected by haptic information used to carry out a visual-oriented matching activity. However, some research has suggested that tactile stimulation suppressed the visual perceptual functions such as visual orientation and discrimination, and this tactile suppression effects primarily arose when the visual and tactile information were temporally and spatially compatible.

Our study demonstrated that the HTP group improved overall visual perceptual functions and had the most significant gains on visual spatial relations, visual memory and visual sequential memory domains. HPT is beneficial for improving visual spatial relations since typically-developing individuals acquire and form spatial concepts through touching and manipulating the objects in their environment. The relationship between haptic perceptions and memory could be supported from the findings that memory is kept and stored in the same neural systems taking part in sensory information processing [35]. Previous findings suggested that components in the somatosensory cortex respond to visual stimulation related to tactile information, and some neurons are part of an active short-term memory and might as well be as involved in cross-modal memory networks [36]. In addition to active short-term memory for tactile stimuli, the cortical somatosensory system might be responsible for constructing long-term memory networks symbolizing stimuli of tactile associated modalities. Haptic perception is strengthened by graded-difficulty matching and exploratory games implemented in haptic modality. No visual, intermodal, or cross-modal information transfer is required, and accordingly, HPT could be also appropriate for training fine motor control difficulties in individuals with visual impairments or visual perceptual dysfunctions.

Individuals with DCD are prone to be affected by certain disruptions in their social participation due to their compromised physical and psychosocial functioning. Tactile perception and component element of haptics are strongly related to behavior regulation and psychosocial development [37]. Sensory modulation disorders such as tactile hypersensitivity are not uncommonly found in individuals with DCD and these adversely affect their psychological adaptation and overall participatory activities [38]. Not only did visual perceptual and fine motor functions benefit from the haptic perceptual training programs in the current study, but the subjects showed improvement on overall adaptive behavior measures consisting of socialization and communication tasks as well. The HPT might be used as an effective intervention to enhance psychosocial functions in DCD adolescents for active participation, and further study was warranted.

Interventions in the form of a home program are always a major part of rehabilitation services, and goal-directed home programs have proven effective in improving functional performance in individuals with disabilities [39]. The HPT proposed in this study is highly reproducible and could provide grounds for developing a home-based intervention while considering both the influence of the pandemic on face-to-face rehabilitation services and expenses for telehealth therapy. This study used such commonly-found objects to design the haptic perceptual training program kits, and it thus could be used as an inexpensive and convenient therapy modality for home or school-based training programs. Parents or school teachers could create their own training kit effortlessly and just simply follow the training procedure in the present study [40,41].

## 5. Conclusions

In conclusion, the HTP conducted on a regular basis was beneficial in improving fine motor control functions in DCD adolescents. More effort should be made to help these participants generalize the training effects to the functional tasks that demand similar motor skills. There are some limitations of the present study. First, the participants could not be randomly assigned because of some practical reasons (e.g., time of the sessions). However, the estimation of intervention effects is less likely to be biased because no differences in age, gender, or preintervention performance were found between the participants in each group. Second, fine motor control might differ in both degree and significance of correlation according to hand dominance. Future studies should distinguish between dominant and non-dominant hands when investigating the effect of haptic perception on fine motor control abilities.

## Figures and Tables

**Figure 1 jcm-11-04755-f001:**
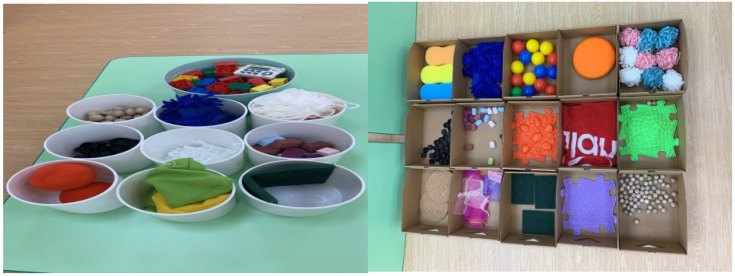
Examples of object sets of the haptic perception training program.

**Table 1 jcm-11-04755-t001:** Group difference at post-intervention.

Test	Group Mean		*p*
	HTP	SOT	
BOT-2			
Fine Motor Precision	13.44 (2.44)	10.89 (3.89)	0.01 *
Fine Motor Integration	13.90 (2.18)	12.65 (2.56)	0.05 *
Manual Dexterity	15.07 (3.07)	11.87 (1.78)	0.01 *
Upper-Limb Coordination	11.99 (2.99)	10.75 (2.57)	0.06
JTFHT			
Objects-dominant hand	33.13 (5.56)	44.10 (6.36)	0.02 *
Object Non-dominant hand	55.08 (6.11)	57.79 (7.38)	0.06
Writing	32.95 (4.98)	50.74 (5.57)	0.04 *
TVPS-4			
Visual Discrimination	13.05 (3.56)	10.11 (2.69)	0.01 *
Visual Memory	14.44 (2.98)	12.20 (2.58)	0.04 *
Visual Spatial Relations	14.15 (2.58)	11.37 (2.17)	0.03 *
Visual Form Constancy	11.53 (1.66)	10.24 (2.89)	0.05 *
Visual Closure	12.05 (1.98)	10.37 (1.46)	0.05
Visual Figure Ground	12.12 (2.22)	10.08 (1.79)	0.04 *
Visual Sequential Memory	13.97 (3.11)	11.82 (1.88)	0.04 *
VABS-C			
Communication	67.50 (10.33)	62.03 (11.98)	0.03 *
Daily Living Skills	60.19 (15.12)	54.63 (63.78)	0.04 *
Socialization Skills	73.31 (14.89)	60.15 (42.28)	0.02 *
Motor Skills	68.87 (11.63)	61.22 (7.99)	0.04 *

BOT-2, Bruininks–Oseretsky Test of Motor Proficiency-Second Edition; JTHFT, Jebsen-Taylor Hand Function Test; TVPS-4, Test of Visual-Perceptual Skills—Fourth Edition; VABS-C, Vineland Adaptive Behavior Scale-Chinese Version. * *p* < 0.05

**Table 2 jcm-11-04755-t002:** Intervention gains within HTP group.

Measurement	Mean		*t*	*p*
	Pre-Test	Post-Test		
BOT-2				
Fine Motor Precision	10.75	13.44	7.33	0.01 *
Fine Motor Integration	10.97	13.18	7.26	0.01 *
Manual Dexterity	12.03	15.07	8.98	0.01 *
Upper-Limb Coordination	10.09	11.99	3.87	0.06
JTFHT				
Objects-dominant hand	41.77	33.13	4.46	0.02 *
Object Non-dominant hand	56.24	55.08	0.89	0.17
Writing	45.88	32.95	6.53	0.04 *
TVPS-4				
Visual Discrimination	10.97	13.05	8.33	0.01 *
Visual Memory	11.03	14.44	9.26	0.01 *
Visual Spatial Relations	11.17	14.15	10.98	0.01 *
Visual Form Constancy	11.02	12.53	2.87	0.05 *
Visual Closure	10.77	11.05	1.28	0.07
Visual Figure Ground	10.23	12.12	1.46	0.06
Visual Sequential Memory	10.24	13.97	0.89	0.17
VABS-C				
Communication	61.11	67.50	5.88	0.02 *
Daily Living Skills	52.15	60.19	5.17	0.03 *
Socialization Skills	65.19	73.31	5.35	0.02 *
Motor Skills	62.69	68.87	4.77	0.04 *

BOT-2, Bruininks–Oseretsky Test of Motor Proficiency-Second Edition; JTHFT, Jebsen-Taylor Hand Function Test; TVPS-4, Test of Visual-Perceptual Skills- Fourth Edition; VABS-C, Vineland Adaptive Behavior Scale-Chinese Version. * *p* < 0.05.

**Table 3 jcm-11-04755-t003:** Intervention gains for HTP and SOT groups.

Measures	HTP		SOT	
	Change	Cohen’s *d*	Change	Cohen’s *d*
BOT-2				
Fine Motor Precision	2.69	0.82 ^a^	2.36	0.20 ^c^
Fine Motor Integration	2.21	0.85 ^a^	1.93	0.61 ^b^
Manual Dexterity	3.04	0.84 ^a^	1.87	0.60 ^b^
Upper-Limb Coordination	1.90	0.61 ^b^	0.55	0.43 ^c^
JTFHT				
Objects-dominant hand	−8.64	0.51 ^b^	−2.43	0.18
Object Non-dominant hand	−1.16	0.12	−2.05	0.11
Writing	−12.93	0.89 ^a^	−1.89	0.23 ^c^
TVPS-4				
Visual Discrimination	2.28	0.74 ^b^	1.84	0.14
Visual Memory	3.41	0.81 ^a^	1.11	0.13
Visual Spatial Relations	2.98	0.85 ^a^	1.42	0.24 ^c^
Visual Form Constancy	0.51	0.18	0.44	0.15
Visual Closure	1.28	0.19	1.42	0.13
Visual Figure Ground	1.89	0.15	1.13	0.12
Visual Sequential Memory	3.73	0.83 ^a^	2.01	0.19
VABS-C				
Communication	6.39	0.82 ^a^	3.88	0.53 ^b^
Daily Living Skills	8.04	0.58 ^b^	5.15	0.57 ^b^
Socialization Skills	8.12	0.60 ^b^	5.19	0.50 ^b^
Motor Skills	6.18	0.78 ^b^	1.77	0.16 ^c^

BOT-2, Bruininks–Oseretsky Test of Motor Proficiency-Second Edition; JTHFT, Jebsen-Taylor Hand Function Test; TVPS-4, Test of Visual-Perceptual Skills- Fourth Edition; VABS-C, Vineland Adaptive Behavior Scale-Chinese Version. ^a^ A Cohen’s *d* > 0.8: large effect size. ^b^ A Cohen’s *d* ≥ 0.5 < 0.8: medium effect size. ^c^ A Cohen’s *d* ≥ 0.2 < 0.5: small effect size.

## Data Availability

The data presented in this study are available on request from the corresponding author. The data are not publicly available due to privacy or ethical restrictions.

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
