# Peer review of "Haptic Perception Training Programs on Fine Motor Control in Adolescents with Developmental Coordination Disorder: A Preliminary Study"

_jcm, 2022, doi:10.3390/jcm11164755_

Round 1
Reviewer 1 Report
Thank you for writing a well-organized paper. I think this is an clinically valuable study.
In page 5, there is a typo; "therapist started the procedure"
Is this a preliminary study? (2. 1 study design). I think it's good to add direction of future study which need more samples, in discussion section.
To improve more scientific accuracy, reinforcement of clinical validity is very helpful. "How neuromuscular disorders were excluded in study sample"
Author Response
Response to Reviewer 1 Comments
Thank you very much for your valuable comments to help us further improve our manuscript. Our revisions in response to your suggestions were described below. And all the revisions were highlighted in the manuscript as well.
Point 1: In page 5, there is a typo; "therapist started the procedure"
Response 1: Thank you. The typo wad corrected.
Point 2:Is this a preliminary study? (2. 1 study design). I think it's good to add direction of future study which need more samples, in discussion section.
Response 2: Thank you. This study was a preliminary study since it was quasi-experimental, and we added the study limitation in the Conclusion section.
Point 3:To improve more scientific accuracy, reinforcement of clinical validity is very helpful. "How neuromuscular disorders were excluded in study sample"
Response 3: The participants were asked to fill out demographic data before starting this study and those who had previous history of neuromuscular disorders or other coexisting disabilities were excluded. This information was added in the Participants section.
Reviewer 2 Report
THIS IS AN INTERESTING PAPER ADDRESSING A RELEVANT ISSUE. IN MY OPINION IT HAS TWO WEAKNESSES. THE FIRST, WHICH CANNOT BE AMENDED, BUT SHOULD BE THROUGHLY DISCUSSED, IS WHY THE TWO INTERVENTION GROUPS WERE NOT RANDOMIZED AND WHICH CRITERIA WERE USED TO ASSINGN PARTICIPANTS TO EITHER GROUP. THE OTHER ACTUALLY QUITE AMENDABLE IF THE AUTHORS AGREE, IS THE DISCREPANCY I NOTICE BETWEEN THE FIRST PART OF THE MANUSCRIPT, WHICH IS SYNTHETICALLY AND CLEARLY PRESENTED, AND THE DISCUSSION/CONCLUSION SECTIONS, WHICH, IN MY OPINION, NEED TO BE REARRANGED AND BETTER ARGUMENTED.
BELOW MY POINT BY POINT COMMENTS:
LN 92 participants were consented to enrollment and publication. WAS IT A WRITTEN CONSENT? DID BOTH PARENTS AND PARTICIPANTS SIGN? PLEASE SPECIFY
LN 154 Eighty eight participants were assigned to two equal-sized (n= 44) groups. THIS IN MY OPINION IS THE MAIN WEAKNESS OF THE STUDY AND SHOULD BE THROUGLTY DISCUSSED AS A STUDY LIMITATION. WHY PARTICIPANTS WERE NOT RANDOMIZED? HAW WAS THE ASSIGNEMENT TO EITHER GROUP PERFORMED?
155 Each intervention group was treated with a 30-minute session every week for 24 156 weeks. IS THERE A RATIO FOR THIS PROTOCOL SUCHE AS REFERENCES FROM SIMILAR STUDIES AND OR PRELIMINARY DATA ON YOUR SIDE?
LN 184 therapist started the procedure REPEATED TWICE
LN 186 VISIONE WAS NOT OCCLUDED IN THE WARM UP TASK?
LN 215-6 Treatment fidelity was verified by examining twenty synchronous online intervention sessions from 2 participating therapists at the first week and 12 weeks: THE SESSIONS WERE RECORDED ONLINE TO THE PURPOSE OF VERYFYING TRATMENT FIDELITY? MAYBE THIS SHOULD BE STATED EARLIER
219 advised activity checklist. A 10-point scale was used, and the av- 219 erage adherence scores of HP . HOW WAS THE CHECKLIST DEVELOPED? I BELIEVE IT SHOULD BE ELABORATED, AND THE CHECKLIST DISPLAYED
LN 298 4. Discussion OVERALL I BELIEVE THE DISCUSSION IS TOO LONG AND A BIT TOO ERRATIC. I WOULD SUGGEST TO REORGANIZE IT AND SYNTHETIZE IT. SOME SUGGESTIONS PRESENTED BELOW:
306 Mature adolescents could explore objects by haptic perception only and are capable 306 of producing various systematic hand movements by different task goals. However I WOULD SUGGEST REPHRASE AS SO: WHILE Mature adolescents could explore objects by haptic perception only and are capable of producing various systematic hand movements by different task goals, PREVIOUS….
LN 306-313 Mature adolescents could explore objects by haptic perception only and are capable 306 of producing various systematic hand movements by different task goals. However, pre- 307 vious studies have suggested that DCD individuals fail to obtain the necessary sensory 308 input and lack the required haptic perceptions to carry out functional hand movements. 13 309 The DCD group performed slower on most hand function tests and daily activities de- 310 manding fine dexterity and with greater difficulty. 28 Given the evidence that fine motor 311 control during purposeful activity significantly relies on constant haptic information in- 312 flow to lead the direction, force, and accuracy of movement, it could be useful clinically 313 to utilize haptic training programs to enhance fine motor functions. MAYBE THIS COULD BE SYNTHETIZED AND MOVED UP IN THE INTRODUCTION
LN 348-351 Consequently, visual perception does not seem to occur in a single module and tends 348 to dynamically interact with other sensory modalities in a myriad of dimensions. Multi- 349 sensory enhancements could happen even the extra-modal input does not provide direct 350 information significant for performing certain tasks. All these findings indicated that 351 training with multiple correlated sensory information is more practical and effective for 352 learning visual perceptual tasks. Haptic perception is strengthened by graded- difficulty 353 matching and exploratory games implemented in haptic modality, there is no visual, in- 354 termodal, or cross-modal information transfer is required, and accordingly, HPT could be 355 also appropriate for training fine motor control difficulties in individuals with visual im- 356 pairments or visual perceptual dysfunctions. OVERALL THIS PART ODF THE DISCUSSION IS NOT CLEAR TO ME: IT SEEMS TO ME THAT GIVEN THE PREMISES A MULTISENSOR INTERVENTION INCLUDING VISUAL CLUES, SUCH AS SOT, SHOULD PROVIDE BETTER RESULTA THAN APTINS TRAINING. WHY DO YPU BELIEVE THIS DOES NOT HAPPEN?
LN366 As a result, HPT could be 366 used as an effective intervention to enhance psychosocial functions in DCD adolescents 367 for active participation. SINCE THIS IS JUST ONE QUASI EXPERIMENTAL STUDY I WOULD REPHRASE THIS SENTENCE WITH A MORA CAUTIOUS STATEMENT
LN 369-379 Although occupational and physical therapy have been recommended in providing 369 both acute and chronic care for those with COVID-19, therapy service offered for individuals with disabilities has been greatly affected. Data during the pandemic showed that many families of individuals with disabilities had decreased access to needed therapy including early intervention, school-based services and outpatient therapies.35 Assessment and intervention have to be modified by adopting alternative services delivery models to ensure continuum of care during the time of restricted access to rehabilitation services. Although telehealth is useful to provide health care when individuals cannot attend an on-site therapy during the pandemic, access to telehealth is inequitable since the devices required for telehealth might not be available for most families. 36 Moreover, telehealth services are not readily useful for rehabilitation disciplines that uniquely rely on direct assessment and intervention. I BELIEVE THIS PARAGRAPH COULD BE DELETED. WE INTRODUCE TELEHEALTH JUST TO SAY IT IS EXPENSIVE, UNEQUITABLE AND NOT APPRORIATE. I BELIEVE THE FURTHER SENTENCE MLING REFERENCE TO TELEHEALTH COULD BE ENOUGH
LN 383 A home-based model of delivery of care could be utilized in the current scenario. THIS ALSO IS INTRODUCED QUITE ABRUPTLY. ACTUALLY THIS STUDY DOES NOT PRESENT A HOME BASED TRAINING AND THERE IS NO WAY TO SAY THAT UNEXPERIENCED UNPROFESSIONAL TRAINERS WOULD OBTAIN THE SAME RESULTS. I WOUDL SUGGEST JUST A COUPLE OF SENTENCES SUGGESTING THAT THIS TRAININS IS HIGHLY REPRODUCIBLE AND THAT COULD PROVIDE GROUND FOR THE DEVELPEMENT OF A HOME BASE INTERVENTION
LN 593 5. Conclusions THE CONCLUSIONS SHOULD JUST SUMMARIZE STUDT RESULTS AND MAY BE HINT SOME CLINICAL APPLICATION/FURTHER DEVELOPEMENT. THIS PARAGRAPH INCLUDES MUCH INFORMATION THAT SHOULD BE REPORTED ELSEWHERE. THE WHOLE CONCLUSION ABOUT HOME BASED TREATMENT IS UNGROUNDED SINCE THE PROPOSED INTEVENTION IS NON A HOME BASED TRAINING.
LN 395 This haptic training program could be implemented easily at home or school during the pandemic period. I SUGGEST REMOVING THIS SENTENCE OF REPHRASING IT AS A POSSIBLE FURTHER STUDY DEVELOPEMENT
LN 397-401 Fine motor control has been assessed by JTHFT in previous studies, and performance time (speed) was the criterion for assessing the hand functioning level. The use of this sole criterion might not be sufficient because haptic perceptual deficits affect not only the speed of performance but as well as the force, direction and accuracy of movement.38 The BOT-2 fine motor tasks were used to provide a complete description 401 of fine motor control performance in the current study. THIS SHOULD BE MOVED UP IN THE DISCUSSION SECTION WHILE DISCUSSING OUTCOME MEASURES OR STRENGTHS OF THE STUDY
LN 403-405 There are some limitations of the present study. Firstly, fine motor control might 403 differ in both degree and significance of correlation according to hand dominance. 404 Future studies should distinguish between dominant and non-dominant hands while in- 405 vestigating the effect of haptic perception on fine motor control abilities THIS ALSO SHOULD BE MOVED UP IN THE DISCUSSION SECTION
LN 406-9 Secondly, adherence to home-based programs has not been evaluated extensively. Home-based models work best when there is regular coaching, follow-up, and feedback from the therapist. The home-based adherence App could be supplemented for supporting such haptic training programs in the future. AGAIN SINCE AS FAR AS I UNDERSTAND THIS IS NOT A HOME BASED INTERVENTION I DO NOT UNDERSTAND THIS PARAGRAPH AND I BELEIVE IT SHOULD BE REMOVED
Author Response
Response to Reviewer 2 Comments
Thank you very much for your valuable comments to help us further improve our manuscript. Our revisions in response to your suggestions were described below. All the revisions were highlighted in the manuscript as well.
Point 1: LN 92 participants were consented to enrollment and publication. WAS IT A WRITTEN CONSENT? DID BOTH PARENTS AND PARTICIPANTS SIGN? PLEASE SPECIFY
Response 1: Yes, it was a written consent, and both parents and participants signed and agreed to participate and publication. We added this description on the Participant section.
Point 2: LN 154 Eighty eight participants were assigned to two equal-sized (n= 44) groups. THIS IN MY OPINION IS THE MAIN WEAKNESS OF THE STUDY AND SHOULD BE THROUGLTY DISCUSSED AS A STUDY LIMITATION. WHY PARTICIPANTS WERE NOT RANDOMIZED? How WAS THE ASSIGNEMENT TO EITHER GROUP PERFORMED?
Response 2: Thank you. All 88 children were randomly assigned to two equal-sized intervention (n= 44) groups by using a computer- generated random table at first, however, nineteen participants (21.6%) asked to be transferred to different intervention group due to their schedule conflicting. Therefore, the participants could not be “truly” randomly assigned because of some practical reasons (e.g., time of the sessions). However, the estimation of intervention effects is less likely to be biased because no differences in age, gender, or preintervention performance were found between the participants in each group. We thoroughly discuss as a study limitation in the Conclusion section.
Point 3: 155 Each intervention group was treated with a 30-minute session every week for 24 weeks. IS THERE A RATIO FOR THIS PROTOCOL SUCH AS REFERENCES FROM SIMILAR STUDIES AND OR PRELIMINARY DATA ON YOUR SIDE?
Response 3: Thank you. The ratios for this protocol are based on the: (1) similar intervention protocols for other developmental disabilities in our previous studies; (2) effectiveness studies of the interventions used for this study. And we added some of the references both in the text and in the References section.
Point 4: LN 184 therapist started the procedure REPEATED TWICE
Response 4: Thank you; it was deleted.
Point 5: LN 186 VISIONE WAS NOT OCCLUDED IN THE WARM UP TASK?
Response 5: Vision was occluded for each participant, and we added this in the text.
Point 6: LN 215-6 Treatment fidelity was verified by examining twenty synchronous online intervention sessions from 2 participating therapists at the first week and 12 weeks: THE SESSIONS WERE RECORDED ONLINE TO THE PURPOSE OF VERYFYING TRATMENT FIDELITY? MAYBE THIS SHOULD BE STATED EARLIER
Response 6: Thank you, we moved up this statement in the Procedure section.
Point 7: 219 advised activity checklist. A 10-point scale was used, and the average adherence scores of HP . HOW WAS THE CHECKLIST DEVELOPED? I BELIEVE IT SHOULD BE ELABORATED, AND THE CHECKLIST DISPLAYED
Response 7: Thank you, we added more detailed description of the checklist on the second paragraph in the Procedure section.
Point 8: 306 Mature adolescents could explore objects by haptic perception only and are capable of producing various systematic hand movements by different task goals. However I WOULD SUGGEST REPHRASE AS SO: WHILE Mature adolescents could explore objects by haptic perception only and are capable of producing various systematic hand movements by different task goals, PREVIOUS….
Response 8: Thank you for your precious suggestion, and the sentence was rephrased accordingly.
Point 9: LN 306-313 Mature adolescents could explore objects by haptic perception only and are capable of producing various systematic hand movements by different task goals. However, previous studies have suggested that DCD individuals fail to obtain the necessary sensory input and lack the required haptic perceptions to carry out functional hand movements. The DCD group performed slower on most hand function tests and daily activities demanding fine dexterity and with greater difficulty. Given the evidence that fine motor control during purposeful activity significantly relies on constant haptic information inflow to lead the direction, force, and accuracy of movement, it could be useful clinically to utilize haptic training programs to enhance fine motor functions. MAYBE THIS COULD BE SYNTHETIZED AND MOVED UP IN THE INTRODUCTION
Response 9: This paragraph was resynthesized and kept in the original section since it was related to the clinical applications of the study results.
Point 10: LN 348-351 Consequently, visual perception does not seem to occur in a single module and tends to dynamically interact with other sensory modalities in a myriad of dimensions. Multi- sensory enhancements could happen even the extra-modal input does not provide direct information significant for performing certain tasks. All these findings indicated that training with multiple correlated sensory information is more practical and effective for learning visual perceptual tasks. Haptic perception is strengthened by graded- difficulty matching and exploratory games implemented in haptic modality, there is no visual, intermodal, or cross-modal information transfer is required, and accordingly, HPT could be also appropriate for training fine motor control difficulties in individuals with visual impairments or visual perceptual dysfunctions. OVERALL THIS PART OF THE DISCUSSION IS NOT CLEAR TO ME: IT SEEMS TO ME THAT GIVEN THE PREMISES A MULTISENSOR INTERVENTION INCLUDING VISUAL CLUES, SUCH AS SOT, SHOULD PROVIDE BETTER RESULT THAN HTP TRAINING. WHY DO YOU BELIEVE THIS DOES NOT HAPPEN?
Response 10: Thank you. We deleted some irrelevant statements and merged into the previous paragraph to make it clearer.
Point 11: LN366 As a result, HPT could be used as an effective intervention to enhance psychosocial functions in DCD adolescents 367 for active participation. SINCE THIS IS JUST ONE QUASI EXPERIMENTAL STUDY I WOULD REPHRASE THIS SENTENCE WITH A MORE CAUTIOUS STATEMENT
Response 11: Thank you. We rephrased this sentence as: “The HPT might be used as an effective intervention to enhance psychosocial functions in DCD adolescents for active participation, and further study was warranted.”
Point 12: LN 369-379 Although occupational and physical therapy have been recommended in providing both acute and chronic care for those with COVID-19, therapy service offered for individuals with disabilities has been greatly affected. Data during the pandemic showed that many families of individuals with disabilities had decreased access to needed therapy including early intervention, school-based services and outpatient therapies. Assessment and intervention have to be modified by adopting alternative services delivery models to ensure continuum of care during the time of restricted access to rehabilitation services. Although telehealth is useful to provide health care when individuals cannot attend an on-site therapy during the pandemic, access to telehealth is inequitable since the devices required for telehealth might not be available for most families. 36 Moreover, telehealth services are not readily useful for rehabilitation disciplines that uniquely rely on direct assessment and intervention. I BELIEVE THIS PARAGRAPH COULD BE DELETED. WE INTRODUCE TELEHEALTH JUST TO SAY IT IS EXPENSIVE, UNEQUITABLE AND NOT APPRORIATE. I BELIEVE THE FURTHER SENTENCE MLING REFERENCE TO TELEHEALTH COULD BE ENOUGH
Response 12: Thank you; we deleted the redundant paragraph accordingly.
Point 13: LN 383 A home-based model of delivery of care could be utilized in the current scenario. THIS ALSO IS INTRODUCED QUITE ABRUPTLY. ACTUALLY THIS STUDY DOES NOT PRESENT A HOME BASED TRAINING AND THERE IS NO WAY TO SAY THAT UNEXPERIENCED UNPROFESSIONAL TRAINERS WOULD OBTAIN THE SAME RESULTS. I WOUDL SUGGEST JUST A COUPLE OF SENTENCES SUGGESTING THAT THIS TRAININS IS HIGHLY REPRODUCIBLE AND THAT COULD PROVIDE GROUND FOR THE DEVELPEMENT OF A HOME BASE INTERVENTION
Response 13: Thank you for the suggestion, and the paragraph was revised.
Point 14: LN 593 Conclusions THE CONCLUSIONS SHOULD JUST SUMMARIZE STUDTY RESULTS AND MAY BE HINT SOME CLINICAL APPLICATION/FURTHER DEVELOPEMENT. THIS PARAGRAPH INCLUDES MUCH INFORMATION THAT SHOULD BE REPORTED ELSEWHERE. THE WHOLE CONCLUSION ABOUT HOME BASED TREATMENT IS UNGROUNDED SINCE THE PROPOSED INTEVENTION IS NON A HOME BASED TRAINING.
Response 14: The Conclusion section was rewritten, some information was removed into the Discussion section and statements related to home-based treatment were deleted.
Point 15: LN 395 This haptic training program could be implemented easily at home or school during the pandemic period. I SUGGEST REMOVING THIS SENTENCE OF REPHRASING IT AS A POSSIBLE FURTHER STUDY DEVELOPEMENT
Response 15: Thank you, we removed this sentence and the following statements related to home-based intervention.
Point 16: LN 397-401 Fine motor control has been assessed by JTHFT in previous studies, and performance time (speed) was the criterion for assessing the hand functioning level. The use of this sole criterion might not be sufficient because haptic perceptual deficits affect not only the speed of performance but as well as the force, direction and accuracy of movement.38 The BOT-2 fine motor tasks were used to provide a complete description 401 of fine motor control performance in the current study. THIS SHOULD BE MOVED UP IN THE DISCUSSION SECTION WHILE DISCUSSING OUTCOME MEASURES OR STRENGTHS OF THE STUDY
Response 16: Thank you, we moved up the paragraphs to the Discussion section.
Point 17: LN 403-405 There are some limitations of the present study. Firstly, fine motor control might differ in both degree and significance of correlation according to hand dominance.Future studies should distinguish between dominant and non-dominant hands while investigating the effect of haptic perception on fine motor control abilities THIS ALSO SHOULD BE MOVED UP IN THE DISCUSSION SECTION
Response 17: Thank you. We still discussed it as a study limitation for providing suggestions for further study.
Point 18: LN 406-9 Secondly, adherence to home-based programs has not been evaluated extensively. Home-based models work best when there is regular coaching, follow-up, and feedback from the therapist. The home-based adherence App could be supplemented for supporting such haptic training programs in the future. AGAIN SINCE AS FAR AS I UNDERSTAND THIS IS NOT A HOME BASED INTERVENTION I DO NOT UNDERSTAND THIS PARAGRAPH AND I BELEIVE IT SHOULD BE REMOVED
Response 18: Thank you, this paragraph was removed.